# Relativistic Two-Photon Matrix Elements for Attosecond Delays

**Jimmy Vinbladh [1,†], Jan Marcus Dahlström [2,*,†] and Eva Lindroth [1,*,†]**

1 Department of Physics, Stockholm University, AlbaNova University Center, SE-106 91 Stockholm, Sweden; jimmy.vinbladh@matfys.lth.se

2 Department of Physics, Lund University, Box 118, SE-221 00 Lund, Sweden

* Correspondence: marcus.dahlstrom@matfys.lth.se (J.M.D.); eva.lindroth@fysik.su.se (E.L.)

† These authors contributed equally to this work.

**Abstract:** The theory of one-photon ionization and two-photon above-threshold ionization is formulated for applications to heavy atoms in attosecond science by using Dirac–Fock formalism. A direct comparison of Wigner–Smith–Eisenbud delays for photoionization is made with delays from the Reconstruction of Attosecond Beating By Interference of Two-photon Transitions (RABBIT) method. Photoionization by an attosecond pulse train, consisting of monochromatic fields in the extreme ultraviolet range, is computed with many-body effects at the level of the relativistic random phase approximation (RRPA). Subsequent absorption and emission processes of infrared laser photons in RABBIT are evaluated by using static ionic potentials as well as asymptotic properties of relativistic Coulomb functions. As expected, light elements, such as argon, show negligible relativistic effects, whereas heavier elements, such a krypton and xenon, exhibit delays that depend on the fine-structure of the ionic target. The relativistic effects are notably close to ionization thresholds and Cooper minima with differences in fine-structure delays predicted to be as large as tens of attoseconds. The separability of relativistic RABBIT delays into a Wigner–Smith–Eisenbud delay and a universal continuum–continuum delay is studied with reasonable separability found for photoelectrons emitted along the laser polarization axis in agreement with prior non-relativistic results.

**Keywords:** attoscience; attophysics; photoionization; above-threshold ionization; Wigner–Smith–Eisenbud delay; Dirac–Fock; RRPA; RABBIT; krypton; xenon

## 1. Introduction

The study of attosecond photoionization dynamics has been made possible by coherent light sources in the extreme ultraviolet (XUV) regime based on high-order harmonic generation (HHG) [1]. Experimental techniques that were originally used for pulse characterization, such as the Reconstruction of Attosecond Beating By Interference of Two-photon Transitions (RABBIT) [2] and the attosecond streak-camera [3], have proved useful to gain novel insights into the time it takes for electrons to escape the binding potentials of atoms [4–16], molecules [17–21], and solid-state targets [22–24]. The main observables are delay-dependent modulations in the photoelectron spectra that arise due a phase-locked laser probe field in the infrared (IR) regime [25–30]. For "weak" fields, these modulations can be understood from perturbation theory, where absorption of one XUV photon ($\Omega$) is followed by exchange of one IR photon ($\pm\omega$). It is a rather technical task to evaluate laser-driven continuum–continuum transitions numerically in the presence of the long-range Coulombic potential: $k' \to k$ [26,31,32]. Thus, analytical continuum–continuum phase shifts $\phi_{cc}(k, k')$, have been derived by using the Wentzel–Kramers–Brillouin (WKB) approximation, in order to interpret the RABBIT delays at sufficiently high kinetic energy of the photoelectrons [33]. Asymptotic theories based on the Eikonal Volkov Approximation (EVA) have also been developed [34]. The main result of these asymptotic theories is that delays observed in RABBIT experiments can be separated into two terms: (i) a finite-difference approximation to the Wigner–Smith–Eisenbud delay of the photoelectron

after absorption of one XUV photon: $\tau_W$ [35–37], and (ii) a universal continuum–continuum delay: $\tau_{cc}(k; \omega)$, with an analytical expression that only depends on the final momentum of the photoelectron and the frequency of the IR field. In the case of a single angular momentum channel $\lambda = \ell_i + 1$ with $\ell_i$ being the initial angular momentum, this separation has been successfully implemented to measure the Wigner-like delay of the $2s$-orbital in neon atoms [12]. In the more general case, where multiple intermediate angular momenta are populated, $\lambda = \ell_i \pm 1$, the probe process becomes more complicated and care must be taken to account for the weight of all intermediate transitions, which leads to an "effective" Wigner delay [33]. As an example, the RABBIT delay measured close to the $3p$-Cooper minimum in argon [38] is much reduced in magnitude when photoelectrons are detected over all emission angles, rather than along the polarization axis of the fields [7]. Nonetheless, the asymptotic theory has been extended to interpret delays from molecules, where contributions of multiple partial waves in the initial orbital and the orientation of the target relative to the laser polarization, adds more complexity to the process [21,39]. Although it has been shown that the separability of delays remains valid at high kinetic energies, by using full two-photon matrix elements from time-independent R-matrix theory [40], the target-specific delay in molecules $\tau_{PI}$, cannot be interpreted as a Wigner-like delay, due to interference effects of multiple partial waves in the two-photon transitions [39] and various channel coupling effects [21,40]. The use of full two-photon R-matrix theory [40] is undoubtedly an important milestone in the field of photoionization delays from molecules, which has allowed for quantitative analysis of many recent experiments [18–21].

In the case of atoms, full two-photon matrix elements have been used for a decade to compute delays in photoionization at various levels of Many-Body Perturbation Theory (MBPT) [41–44]. Although the importance of the random-phase approximation with exchange (RPAE) for attosecond science was first realized by Kheifets [45,46], numerical simulations of the one-photon ionization process, developed by Amusia [47], are inherently insufficient to interpret RABBIT delays. Thus, a two-photon approach was developed, whereby the many-body response of XUV absorption was computed at the level of RPAE, and the IR exchange in the continuum was computed numerically by using an effective one-body ionic potential [41,42]. This two-photon matrix approach has shown good agreement with a range of RABBIT experiments [7,8,12,13,48]. Noble gas atoms consist of multiple valence states, which implies experimentally unresolved ionic states with magnetic quantum numbers $|m| \le \ell_i$. However, any problem with incoherent final channels can easily be avoided by detecting photoelectrons along the polarization axis $\hat{z}$, where only $m = 0$ contributes. In this configuration, it has been shown that a numerically obtained continuum–continuum delay, $\tau_{cc}^{MBPT}$, can be accurately separated from the one-photon Wigner delay $\tau_W^{MBPT}$, computed for photoelectrons along the polarization axis with the unique ionic state $m = 0$ [41,43]. In this way, a precise separation of delays has been demonstrated down to 5 eV, which is much lower than the high-energy regime predicted by the original asymptotic theory [33]. The two-photon matrix elements have also been used to study effects beyond the asymptotic approximation. Firstly, a strong angle-dependence of RABBIT delays from the isotropic helium atom was evidenced in experiments by Heuser et al. [48]. Secondly, the role of universal asymmetries between absorption and emission processes in the continuum was identified by Busto et al. [49]. Finally, a weak angular-momentum dependence of continuum–continuum phases was measured by Fuchs et al. in helium atoms [50]. The latter discovery was in good agreement with theoretical predictions based on exact two-photon matrix elements for hydrogen, provided by Taïeb [33], as well as full two-photon matrix elements based on MBPT [13]. Thus, several effects that depend on the exact form of continuum states have been identified by using RABBIT delay measurements in recent years [51].

Due to the energy spacing between the odd harmonics from HHG, $\Delta\Omega = 2\omega$, the temporal resolution in traditional RABBIT experiments is limited to probe processes that are much shorter than $2\pi/\Delta\Omega = T_\omega/2 \approx 1.3$ fs (assuming an IR laser system with $\hbar\omega = 1.55$ eV). At a first glance, this seems to preclude any studies of autoionizing dynamics

in atoms or molecules, which typically unfold on a time scale of a few femtoseconds, or more [29,52]. However, the subject of combined time–frequency non-linear metrology is quite subtle, and it has been found that a high-energy resolution of photoelectrons in RABBIT sidebands can be used to reconstruct autoionizing processes in time [11]. In this case of resonant excitation, via bound Rydberg states or autoionizing states, it is useful to consider the RABBIT scheme as a combination of one "structured" (resonant) path and another "unstructured" (reference) path [10,11,13,16,53,54]. In this case, the phase variation of the resonant path is typically much stronger than any continuum–continuum (or other non-resonant) phase shift, and the phenomena can be understood by expanding Fano's model for autoionization to laser-assisted photoionization, within the strong-field approximation [55], or by using approximate two-photon two-color matrix elements [56,57]. In the latter works, it was shown that finite pulses, in the time domain, can lead to non-periodic structures in RABBIT experiments due to autoionizing states. The two-photon Fano model has proven essential to disentangle dynamics from multiple autoionizing states measured by the RABBIT technique [14]. Although we find that the theory development for autoionization in RABBIT is another milestone in the field, we will not consider this class of processes in the following work. Rather, we will focus on correlation effects in unstructured continuum, where MBPT is a numerically efficient route to describe correlation effects and RABBIT data can be safely assumed to be periodic.

Despite these many successes, there remained disagreement between experimental and theoretical results for the relative RABBIT delay between the $3p$ and $3s$ orbitals in argon, first measured by Klünder et al. in 2011 [5], which was mostly ascribed to the low signal close to the correlated minimum in the $3s$-partial photoionization cross section [6,41,42,58–60]. The fact that this exceptionally deep minimum from $3s$ arises due to correlation effects, was first showed by Amusia in 1972 by applying the RPAE method to describe photoionization from inner atomic orbitals [61]. By using two-photon matrix elements, it has now been shown that the position, height, and sign of the associated RABBIT delay from $3s$ is similarly sensitive to correlation effects [41,42], which largely stems from the sensitivity of the one-photon Wigner delay peak from the correlated minimum in the photoionization cross section [62]. In order to solve this long-standing problem, a full two-photon two-color RPAE (2P2C-RPAE) method was developed for RABBIT delays [44]. This new method allowed for detailed examination of correlated IR exchange processes. It was found that, apart from a rather minor discrepancy at the correlated $3s$-minimum in argon, the universal separability of the MBPT continuum–continuum delay and Wigner delay was achieved. However, this discrepancy was still not enough to reach agreement with the experimental results [5,6]! It was not until the argon experiment was repeated, with higher photon energies by Alexandridi et al. in 2021 [15], that excellent agreement with 2P2C-RPAE results was found in a broad energy range above the $3s$-minimum in argon. It was also concluded that the long-standing $3p$-$3s$ problem was caused by an "accidental" harmonic satellite, namely the $3s^2 3p^4 (^1D) 4p (^2P)$ shake-up process, predicted by Wijesundera and Kelly in 1989 by using MBPT [63], which overlapped with the $3s$-RABBIT sidebands. Prior to that, the importance of "two-electron-two-hole" excitations in argon had been found by Amusia and Kheifets by considering effects beyond RPAE in 1981 [64].

The 2P2C-RPAE method also opened up for gauge-invariance tests of the RABBIT theory [44]. It was concluded that the so-called length-gauge formulation of light-matter interaction was much favoured, which is in line with the gauge theory of Kobe [65,66]. In the velocity-gauge formulation of RABBIT, it was found that the interaction with the second photon required a more detailed many-body treatment, beyond the one-body ionic potential, with important contributions from both time-orders of the fields XUV+IR and IR+XUV. Although it was shown that only the *complete* 2P2C-RPAE theory leads to gauge-invariant results, the approximate one-body treatment of the IR-exchange was shown to be an excellent approximation in length gauge. For this reason, we will use the length gauge, with an effective ionic potential to describe IR exchange processes, in our current work, which aims to quantitatively account for relativistic effects in RABBIT experiments.

The study of relativistic effects is quite a recent development in attosecond physics. In our view, Saha et al. have pioneered this field with calculations of relativistic one-photon Wigner delays [62,67,68], based on the relativistic random phase approximation (RRPA). Although RRPA theory was originally developed in the late seventies by Johnson and Cheng to describe one-photon ionization cross sections in heavy elements [69,70], the interest in such phenomena is revived by recent RABBIT experiments that have targeted heavy elements. First, Jordan et al. [71] and Jain et al. [72] have compared photoelectrons from the fine-structure split valance orbitals: $4p_j$ and $5p_j$ with $j = 1/2$ and $3/2$ of krypton and xenon atoms, respectively, and secondly, Jain et al. [73] and Zhong et al. [74], have compared photoelectrons from inner orbitals in xenon, down to the $4d$ orbital. The $4d$ orbital is of special interest because it is known to posses a giant collective resonance in the photoionization cross section, as evidenced by MBPT in the early seventies by Amusia and Wendin [75,76]. Thus, it is now possible to study the role of sizable relativistic effects, such as the spin-orbit effect in xenon, in the time domain with RABBIT. This opens a call for time-dependent methods to solve the Dirac equation for heavy many-electron atoms; as an example we mention the recently developed relativistic time-dependent configuration–interaction singles (RTDCIS) method [77], but also extend the computation of two-color, two-photon matrix elements to the relativistic domain. Concerning the lack of such relativistic theories, we mention that in refs. [71,73], the experiments were accompanied by photoionization delay calculations with one-photon matrix elements at the level of RRPA for XUV absorption, whereas various asymptotic formulas from non-relativistic theory were used to account for IR exchange effects. Our goal here is to treat the whole process within a relativistic framework and below we discuss the different points where the relativistic treatment differs from that of the non-relativistic one with an effective ionic potential for IR exchange [41–43]. We also mention that the method presented here has already been utilized in various projects, such as [49,74], without any detailed description of the theoretical formulation. A full development of the two-photon, two-color relativistic random phase approximation (2P2C-RRPA) is beyond the scope of the present work, but we expect that it would not lead to any major modification of the results presented here, because we base our entire theory on the length gauge formulation of the light–matter interaction, where the one-body ionic potential description of IR exchange processes is a good approximation [44].

In Section 2 below, some basic concepts are introduced, and the relativistic scattering phases, as well as the asymptotic form of the continuum solutions, are discussed in detail. Section 3 discusses photoionization delay in a relativistic framework, and in Section 4 the many-body implementation is outlined, and the technique to calculate the needed two-photon matrix elements is explained. Some results are finally shown in Section 5.

## 2. Theory

### 2.1. The Dirac Equation

The starting point for calculations in a relativistic framework is the Dirac equation. We aim here for calculations on many-electron systems, and as a first approximation we let the electron–electron interaction be approximated by an average potential: the relativistic version of the Hartee–Fock (HF) potential, usually called the Dirac–Fock (DF) potential. Each electron is then governed by the one-particle Hamiltonian:

$$h_{DF} = c\boldsymbol{\alpha} \cdot \mathbf{p} + \left( u_{DF} - \frac{e^2}{4\pi\varepsilon_0} \frac{Z}{r} \right) \mathbf{1}_4 + mc^2 \beta, \tag{1}$$

with eigenvalues labeled by $E$, and where $\boldsymbol{\alpha}$ is expressed in Pauli matrices and $\beta$ has the corresponding form

$$\boldsymbol{\alpha} = \begin{pmatrix} 0 & \sigma \\ \sigma & 0 \end{pmatrix}, \quad \beta = \begin{pmatrix} \mathbf{1}_2 & 0 \\ 0 & -\mathbf{1}_2 \end{pmatrix}. \tag{2}$$

For closed shell atoms, as the rare gases treated here, the Dirac–Fock potential is spherically symmetric, and the two-component radial part of the wave function can be separated out and determined by the radial Hamiltonian

$$\left(h_\kappa^{DF}(r) - mc^2\right)\begin{pmatrix} f_\kappa(r) \\ g_\kappa(r) \end{pmatrix} = \left(E - mc^2\right)\begin{pmatrix} f_\kappa(r) \\ g_\kappa(r) \end{pmatrix} = \epsilon\begin{pmatrix} f_\kappa(r) \\ g_\kappa(r) \end{pmatrix} \tag{3}$$

with

$$\left(h_\kappa^{DF}(r) - mc^2\right) = \begin{pmatrix} u_{DF}(r) - \frac{e^2}{4\pi\varepsilon_0}\frac{Z}{r} & -c\hbar\left(\frac{d}{dr} - \frac{\kappa}{r}\right) \\ c\hbar\left(\frac{d}{dr} + \frac{\kappa}{r}\right) & u_{DF}(r) - \frac{e^2}{4\pi\varepsilon_0}\frac{Z}{r} - 2mc^2 \end{pmatrix}, \tag{4}$$

where $f_\kappa$ is the upper, typically larger, component, and $g_\kappa$ the lower, typically smaller, component. The four-component eigenfunction to the one-particle Hamiltonian in Equation (1) can now be written as [78]

$$\psi_{n\ell jm}(r,\theta,\phi) = \begin{pmatrix} \frac{f_{n\ell j}(r)}{r}\chi_{\kappa m}(\theta,\phi) \\ \frac{ig_{n\ell j}(r)}{r}\chi_{-\kappa m}(\theta,\phi) \end{pmatrix}$$

$$= \begin{pmatrix} \frac{f_{nj\ell}(r)}{r}\sum_{\nu,\mu}\langle \ell\mu s\nu \mid jm\rangle\xi_\nu Y_{\ell,\mu}(\theta,\phi) \\ \frac{ig_{nj\ell}(r)}{r}\sum_{\nu,\mu}\langle (2j-\ell)\mu s\nu \mid jm\rangle\xi_\nu Y_{(2j-\ell),\mu}(\theta,\phi) \end{pmatrix}, \tag{5}$$

where $\chi_{\kappa m}(\theta,\phi)$ is a vector coupled function of a spherical harmonic and a spin function $\xi_\nu$. The relativistic quantum number $\kappa$ is defined by the eigenvalue equation $(\sigma \cdot \ell + 1)\chi_{\kappa m} = -\kappa\chi_{\kappa m}$ and takes the value $\kappa = \ell(\ell+1) - j(j+1) - 1/4$. When $\kappa$ is negative, $(j = \ell + 1/2)$, the spherical harmonic associated with the small component, will be one unit of orbital angular momenta larger than that for the large component, and vice verse for positive $\kappa$ $(j = \ell - 1/2)$.

The RRPA method, which is also known as the linear response within the time-dependent Dirac–Fock (TDDF) formalism, will be used to describe the atomic response to electromagnetic radiation. It accounts for the interaction with the electromagnetic field in lowest order, including also corrections to the static Dirac–Fock potential by field-perturbed orbitals [47,79]. The method is discussed further in Section 4. In the next section, we will discuss expressions for the radial continuum wave functions at large, but not infinite distances from the ion.

## 2.2. The Scattering Phase of the Photoelectron

Although the total photoionization cross section is determined by the amplitude of the outgoing electron wave packet, its phase is crucial for its angular dependence as well as its delayed appearance in the continuum. In the following, we discuss the difference of the scattering phase in a relativistic formulation compared to the non-relativistic one.

We consider first an $N$-electron atom that absorbs a photon and subsequently ejects a photoelectron from orbital $b$. The radial photoelectron wave function will in the non-relativistic case be described by an outgoing phase-shifted Coulomb wave that asymptotically has the form

$$u_{q,\Omega,b}^{(1)}(r) \approx -\pi M_{\text{nrel}}^{(1)}(q,\Omega,b)\sqrt{\frac{2m}{\pi k\hbar^2}}\, e^{i\left(kr + \frac{Z}{ka_0}\ln 2kr - \ell\frac{\pi}{2} - \sigma_{Z,k,\ell} + \delta_{k,\ell}\right)}. \tag{6}$$

Here energy normalization is assumed, and $M_{\text{nrel}}^{(1)}$ is the non-relativistic electric dipole transition matrix element to the final continuum state $q$ with momenta $k$, $\ell$, and $m$. Although

$M_{\mathrm{nrel}}^{(1)}$ can be chosen to be real in a one-electron context it will be complex when correlation effects are considered. The Coulomb phase is

$$\sigma_{Z,k,\ell} = \arg\left[\Gamma\left(\ell + 1 + \frac{iZ}{ka_0}\right)\right], \tag{7}$$

for a photoelectron in the field from a point charge of $Ze$. Note that in Equations (6) and (7), we use the negative Coulomb phase convention, rather than the equivalent positive sign convention that is more commonly used: cf. Equations (1) and (2) in Ref. [44], in order to easily relate the phase expressions to existing relativistic theory in the literature [80]. The additional phase shift $\delta_{k,\ell}$ comes from the short range many-body potential of the final state. The Bohr radius is here denoted with $a_0$. In the relativistic case, the asymptotic radial wave function will have an upper and a lower component, cf. Equation (5), which will have the form [80]

$$u_{q,\Omega,b}^{(f,1)}(r) \approx -\pi M^{(1)}(q,\Omega,b)\sqrt{\frac{2m}{\pi k\hbar^2}\left(1 + \frac{\epsilon}{2mc^2}\right)} \times e^{i\left(kr + \eta\ln 2kr - \gamma\frac{\pi}{2} - \tilde{\sigma}_{Z,k,\gamma} + \nu + \tilde{\delta}_{Z,k,\gamma}\right)},$$

$$u_{q,\Omega,b}^{(g,1)}(r) \approx -i\zeta\pi M^{(1)}(q,\Omega,b)\sqrt{\frac{2m}{\pi k\hbar^2}\left(1 + \frac{\epsilon}{2mc^2}\right)} \times e^{i\left(kr + \eta\ln 2kr - \gamma\frac{\pi}{2} - \tilde{\sigma}_{Z,k,\gamma} + \nu + \tilde{\delta}_{Z,k,\gamma}\right)}, \tag{8}$$

where the superscripts $f$ and $g$ indicate the large (upper) and small (lower) components respectively and

$$\zeta = \sqrt{\frac{E - mc^2}{E + mc^2}} = \frac{k\hbar}{2mc}\frac{1}{\left(1 + \frac{\epsilon}{2mc^2}\right)} \tag{9}$$

is the relation between the large and small component at infinity. This asymptotic relation is given directly by Equation (4), with $\epsilon = E - mc^2$ being the kinetic energy at infinity. The form of the components in Equation (8) is indeed the same as in the non-relativistic case, but the parameters have slightly changed definition: $M^{(1)}$ is now the relativistic matrix element, and $k$ is calculated from the relativistic kinetic energy as

$$k = \frac{\sqrt{E^2 - m^2c^4}}{\hbar c} = \frac{\sqrt{2\epsilon m}}{\hbar}\sqrt{1 + \frac{\epsilon}{2mc^2}}. \tag{10}$$

The first factor on the right-hand side of Equation (10) is identical to the non-relativistic expression for $k$, which is thus only slightly adjusted as long as the kinetic energy of the released electron is modest: $\epsilon \ll mc^2$. The constant $\eta$ is given by

$$\eta = Z\alpha E\sqrt{\frac{1}{E^2 - m^2c^4}} = \frac{Z}{a_0 k}\left(\frac{\epsilon}{mc^2} + 1\right) \tag{11}$$

where $\alpha$ is the fine structure constant, $\alpha = \hbar/(a_0 mc)$. In the non-relativistic limit $\eta$ will thus tend to $Z/(a_0 k)$ as expected by comparison with Equation (6). The relativistic Coulomb phase is

$$\tilde{\sigma}_{Z,k,\gamma} = \arg[\Gamma(\gamma + i\eta)] \tag{12}$$

with

$$\gamma = \sqrt{\kappa^2 - \alpha^2 Z^2} \tag{13}$$

and

$$\nu = \frac{1}{2} \arg\left[\frac{-\kappa + \frac{iZ}{ka_0}}{\gamma + i\eta}\right]. \tag{14}$$

The phase induced by the short-range part of the many-body potential for the final state is denoted with $\tilde{\delta}_{Z,k,\kappa}$.

### 2.3. Phase-Shifted Relativistic Coulomb Functions at Large Distances

Calculations on many-body systems have to be done numerically. While the wave function for the escaping photoelectron will differ from the analytically known Coulombic ones at short distances, it will approach a combination of a phase-shifted known regular and irregular Coulomb function outside the core of the remaining ion. Because transition matrix elements between continuum states do not converge on a finite grid, it is convenient to have access to continuum solutions, with a possible phase shift $\delta$, that can be used to continue the integration to infinity. We are here interested to find expressions for the relativistic case, but it is illustrative to compare with the more studied non-relativistic formulation.

The solutions to the hydrogen-like Schrödinger equation with positive energy is given by the Coulomb functions (see e.g., [81]). The regular Coulomb function is in particular

$$F_\ell(\eta_{\mathrm{nrel}}, kr) = \frac{1}{2} e^{\frac{\pi}{2}\eta_{\mathrm{nrel}}} \frac{|\Gamma(\ell + 1 + i\eta_{\mathrm{nrel}})|}{(2\ell + 1)!} e^{-ikr} (2kr)^{\ell+1} M(\ell + 1 + i\eta_{\mathrm{nrel}}, 2\ell + 2, 2ikr), \tag{15}$$

where $M$ is the confluent hypergeometric function, $\sigma$ is defined in Equation (7) and $\eta_{\mathrm{nrel}} = Z/(a_0 k)$. Non-relativistic Coulomb functions expressions, valid for large $kr$, are provided in Ref. [82]:

$$F_\ell = \bar{g}\cos\Delta_{\mathrm{nrel}} + \bar{f}\sin\Delta_{\mathrm{nrel}} \tag{16}$$
$$G_\ell = \bar{f}\cos\Delta_{\mathrm{nrel}} - \bar{g}\sin\Delta_{\mathrm{nrel}}, \tag{17}$$

for the regular, $F_\ell$, and irregular, $G_\ell$, Coulomb functions respectively, where

$$\Delta_{\mathrm{nrel}} \equiv kr + \frac{Z}{ka_0}\ln 2kr - \frac{\pi}{2}\ell - \sigma_{Z,k,\ell} + \delta \tag{18}$$

and $\bar{f}$ and $\bar{g}$, which depend on $Z$, $r$, $k$ , and $\ell$, can be obtained through simple recursive formulas given in ref. [82]. When $r \to \infty$, $\bar{g} \to 0$ and $\bar{f} \to 1$ and thus the regular function approaches a sin-function, and the irregular a cos-function, both with amplitude one. The combination

$$F_\ell(\eta_{\mathrm{nrel}}, kr) - iG_\ell(\eta_{\mathrm{nrel}}, kr) \tag{19}$$

will thus asymptotically approach an outgoing wave, with modulus square equal to unity. Energy normalized continuum functions are obtained by multiplications with $\sqrt{2m/\pi k\hbar^2}$.

It is interesting to note that Equations (16) and (17) imply that the irregular (regular) function can readily be obtained when the regular (irregular) one is at hand. In the former case, the irregular solution is found as

$$G_\ell = \frac{\left(\frac{dF_\ell}{dr} - \frac{F_\ell}{\bar{g}^2 + \bar{f}^2}\left(\frac{d\bar{g}}{dr}\bar{g} + \frac{d\bar{f}}{dr}\bar{f}\right)\right)}{k + \eta/r + \frac{1}{\bar{g}^2 + \bar{f}^2}\left(\frac{d\bar{g}}{dr}\bar{f} - \frac{d\bar{f}}{dr}\bar{g}\right)}. \tag{20}$$

Turning to the relativistic Coulomb problem, we set out to find the relativistic counterparts to Equations (16) and (17), which to the best of our knowledge, are not available in the

literature. The exact two-component relativistic regular, $\tilde{F}_\gamma$, and irregular, $\tilde{G}_\gamma$, solutions are given in a pioneering article by Johnson and Cheng [80]. In particular the regular solution is

$$\tilde{F}_\gamma(\eta, kr) = \sqrt{\frac{E + mc^2}{2E}} \frac{1}{2} e^{\frac{\pi}{2}\eta} \frac{|\Gamma(\gamma + i\eta)|}{\Gamma(2\gamma + 1)} (-2ikr)^\gamma e^{ikr}$$
$$\begin{pmatrix} \left(-\kappa + \frac{iZ}{ka_0}\right) M_\gamma + (\gamma - i\eta) M_{\gamma+1} \\ -i\zeta\left(\left(-\kappa + \frac{iZ}{ka_0}\right) M_\gamma - (\gamma - i\eta) M_{\gamma+1}\right) \end{pmatrix} \tag{21}$$

with $\zeta$, $\gamma$, $\eta$ and $k$ given in Equations (9)–(11) and (13), and the short-hand notation

$$\begin{aligned} M_\gamma &= & M(\gamma - i\eta, 2\gamma + 1, -2iz) \\ M_{\gamma+1} &= & M(\gamma + 1 - i\eta, 2\gamma + 1, -2iz) \end{aligned} \tag{22}$$

has been used for the confluent hypergeometric functions.

An asymptotic expansion of the confluent hypergeometric function, $M$ can be found in ref. [83], which indeed can be used to obtain asymptotic expansions for $\tilde{F}_\gamma$ and $\tilde{G}_\gamma$ on forms similar to Equations (16) and (17):

$$\tilde{F}_\gamma = \sqrt{\frac{E + mc^2}{2E}} \begin{pmatrix} \bar{f}_{\text{large}} \cos \Delta - \bar{g}_{\text{large}} \sin \Delta \\ -\zeta(\bar{g}_{\text{small}} \cos \Delta + \bar{f}_{\text{small}} \sin \Delta) \end{pmatrix} \tag{23}$$

and

$$\tilde{G}_\gamma = \sqrt{\frac{E + mc^2}{2E}} \begin{pmatrix} -(\bar{g}_{\text{large}} \cos \Delta + \bar{f}_{\text{large}} \sin \Delta) \\ -\zeta(\bar{f}_{\text{small}} \cos \Delta - \bar{g}_{\text{small}} \sin \Delta) \end{pmatrix} \tag{24}$$

with

$$\Delta = kr + \eta \ln 2kr - \pi\gamma/2 - \tilde{\sigma}_{Z,k,\gamma} + \nu + \tilde{\delta} \tag{25}$$

with $\tilde{\sigma}$ and $\nu$ given in Equations (12) and (14). The possible extra phase shift is denoted by $\tilde{\delta}$. In the non-relativistic limit $\Delta \to \Delta_{\text{nrel}} \pm \pi/2$, for $\kappa > 0$ and $\kappa < 0$ respectively, and thus the sin/cos—functions in Equations (16) and (17) are replaced with $\mp \cos / \pm \sin$ in the upper components of Equation (23) and (24). The relativistic $\bar{f}, \bar{g}$ functions are obtained as

$$\bar{f}_{\text{large/small}} = \text{Re}(\aleph_\pm) \tag{26}$$
$$\bar{g}_{\text{large/small}} = \text{Im}(\aleph_\pm) \tag{27}$$

from

$$\aleph_\pm = \sum_{n=0} \frac{(\gamma - i\eta)_n (-\gamma - i\eta)_n}{n!} (2ikr)^{-n} \pm$$
$$\sum_{n=0} \frac{(\gamma + 1 + i\eta)_n (-\gamma + 1 + i\eta)_n}{n!} (-2ikr)^{-n} \tag{28}$$

where $(a)_n = a(a+1)(a+2)\ldots(a+n-1)$, $(a)_0 = 1$. Similarly to the non-relativistic case $\bar{g}_{\text{large/small}} \to 0$ and $\bar{f}_{\text{large/small}} \to 1$ when $r \to \infty$. Thus the upper regular, and the lower irregular, approach $\cos \Delta$, whereas the lower regular and the upper irregular tend to $\sin \Delta$. The asymptotic expressions are thus

$$\tilde{F}_\gamma(\eta, kr) \to \sqrt{\frac{E + mc^2}{2E}} \begin{pmatrix} \cos \Delta \\ -\zeta \sin \Delta \end{pmatrix}, \tag{29}$$

$$\tilde{G}_\gamma(\eta, kr) \to \sqrt{\frac{E + mc^2}{2E}} \begin{pmatrix} -\sin\Delta \\ -\zeta\cos\Delta \end{pmatrix}, \tag{30}$$

when $kr \to \infty$, and the combination

$$\tilde{F}_\gamma(\eta, kr) - i\tilde{G}_\gamma(\eta, kr) \to \sqrt{\frac{E + mc^2}{2E}} \begin{pmatrix} 1 \\ i\zeta \end{pmatrix} e^{i\Delta} \tag{31}$$

will, in close analogy with the non-relativistic expression in Equation (19), asymptotically approach an outgoing wave, with modulus square unity. The energy normalized functions are again obtained by multiplication with $\sqrt{2m/\pi k\hbar^2}$. We note finally that Equation (20) holds also in a relativistic framework. It provides the irregular solution, $\tilde{G}_\gamma$ from $\tilde{F}_\gamma$, if $\bar{f}$ and $\bar{g}$ are just replaced with $\bar{f}_{\text{large}}$ and $\bar{g}_{\text{large}}$ or $\bar{f}_{\text{small}}$ and $\bar{g}_{\text{small}}$ for the upper and lower components respectively.

## 3. Delay in Photoionization

We will here briefly discuss the calculation of delays in laser-assisted photoionization, emphasizing the differences compared to the non-relativistic description. A detailed account of the latter can be found in refs. [42,44].

### 3.1. The Wigner Delay

The concept of delay was introduced by Wigner [35], Smith [36] and Eisenbud [37] as the derivative of the scattering phase with respect to energy. With a finite difference approximation of the derivative $\Delta\omega = 2\omega$, the Wigner contribution to the atomic delay measured in a RABITT experiemnt is

$$\tau_W = \frac{\phi_> - \phi_<}{2\omega}, \tag{32}$$

where $\phi_{>/<}$ refer to the phases acquired in the XUV absorption step in the two paths where either the higher or the lower harmonic is absorbed. Non-relativistically, and for detection of the photoelectron in the $\hat{z}$ direction, these phases are

$$\phi_>^{\text{nrel}} = \arg\left(\sum_\ell M_>^{\text{nrel}}(\ell) e^{i\left(-\ell\frac{\pi}{2} - \sigma_{Z,k_>,\ell} + \delta_{k_>,\ell}\right)} Y_{\ell,0}(\hat{z})\right)$$

$$\phi_<^{\text{nrel}} = \arg\left(\sum_\ell M_<^{\text{nrel}}(\ell) e^{i\left(-\ell\frac{\pi}{2} - \sigma_{Z,k_<,\ell} + \delta_{k_<,\ell}\right)} Y_{\ell,0}(\hat{z})\right), \tag{33}$$

where the short-hand notation for the one-photon matrix elements, $M_{>/<}(\ell) \equiv M^{(1)}(q_{>/<}, \Omega_{>/<}, b)$, with final photoelectron wave number $k_{>/<}$ and angular momentum $\ell$, after absorption of a photon with angular frequency $\Omega_{>/<}$, is used. Relativistically the corresponding amplitudes have two components and it is more appropriate to define the Wigner delay as

$$\tau_W = \frac{1}{2\omega} \arg\left[\sum_{m=\pm\frac{1}{2}}\right.$$

$$\left(\sum_\kappa M_< \begin{pmatrix} \chi_{\kappa m}(\hat{z}) \\ i\zeta\chi_{-\kappa m}(\hat{z}) \end{pmatrix} e^{i\left(-\gamma\frac{\pi}{2} - \tilde{\sigma}_{Z,k,\gamma} + \nu + \tilde{\delta}_{Z,k,\gamma}\right)}\right)^\dagger$$

$$\left.\left(\sum_{\kappa'} M_> \begin{pmatrix} \chi_{\kappa'm}(\hat{z}) \\ i\zeta\chi_{-\kappa'cm}(\hat{z}) \end{pmatrix} e^{i\left(-\gamma'\frac{\pi}{2} - \tilde{\sigma}_{Z,k,\gamma'} + \nu' + \tilde{\delta}_{Z,k,\gamma'}\right)}\right)\right], \tag{34}$$

where the calculation of the delay of electrons emitted along the z-axis requires an incoherent sum over $m = \pm 1/2$. The two incoherent contributions to the Wigner delay are due to unresolved photoelectron spin in the final state.

### 3.2. The Atomic Delay

We now consider measurements that employ the RABBIT technique [2], where an XUV comb of odd-order harmonics of a fundamental laser field with angular frequency $\omega$, is combined with a synchronized, weak laser field with the same angular frequency. In RABBIT, the one-photon ionization process is assisted by an IR photon that is either absorbed or emitted. The same final state is reached when both an XUV harmonic with energy $\hbar\Omega_< = (2n-1)\hbar\omega$ and an IR photon is absorbed, as when the next XUV harmonic, $\hbar\Omega_> = (2n+1)\hbar\omega$, is absorbed while an IR photon is emitted. This gives rise to modulated sidebands in the photoelectron spectrum at energies corresponding to the absorption of an even number of IR photons. Schematically the intensity of such a sideband can be written as [25]

$$S = \mid A_{\mathrm{a}} + A_{\mathrm{e}} \mid^2 = \mid A_{\mathrm{a}} \mid^2 + \mid A_{\mathrm{e}} \mid^2 + A_{\mathrm{e}}^* A_a + A_{\mathrm{e}} A_{\mathrm{a}}^*$$
$$\mid A_{\mathrm{a}} \mid^2 + \mid A_{\mathrm{e}} \mid^2 + 2 \mid A_{\mathrm{e}} \mid \mid A_{\mathrm{a}} \mid \cos[\arg(A_{\mathrm{e}}) - \arg(A_{\mathrm{a}})], \tag{35}$$

where $A_{\mathrm{a/e}}$ are the complex quantum amplitudes for the two-photon processes involving absorption (a) or emission (e) of an IR photon, and leading to the same final energy. The modulation arises from the last term in Equation (35) and can be shown to be governed by the delay between the IR and XUV pulses, $\tau$, the group delay of the attosecond pulses in the train, $\tau_{XUV}$, and by a contribution from the atomic system which is due the phase difference between the emission and the absorption paths in the atom:

$$\cos[\arg(A_{\mathrm{e}}) - \arg(A_{\mathrm{a}})] = \cos[2\omega(\tau - \tau_{XUV}) + \phi_{\mathrm{e}} - \phi_{\mathrm{a}}]. \tag{36}$$

The atomic contribution can be interpreted as an atomic delay: $\tau_A = (\phi_{\mathrm{e}} - \phi_{\mathrm{a}})/2\omega$. Because the delay between the two light fields is controlled in the experiments and the pulse train group delay can be canceled through relative measurements, the atomic contribution can be extracted. A recent review of the experimental method can be found in [84]. In the following, we discuss the determination of $\phi_{\mathrm{a}}$ and $\phi_{\mathrm{e}}$.

The outgoing radial wave function for the large component, after interaction with two photons, will, in accordance with the one-photon situation in Equation (8), have the asymptotic form

$$u_{q,\omega,\Omega,b}^{(f,2)}(r) \approx -\pi M^{(2)}(q,\omega,\Omega,b)\sqrt{\frac{2m}{\pi k \hbar^2}\left(1 + \frac{\varepsilon}{2mc^2}\right)}\, e^{i\left(kr + \eta ln2kr - \gamma\frac{\pi}{2} - \tilde{\sigma}_{Z,k,\gamma} + \nu + \tilde{\delta}_{Z,k,\gamma}\right)}, \tag{37}$$

where the important difference compared to the one-photon case lies in the presence of the two-photon transition element $M^{(2)}$, which connects the initial state $b$ to the continuum state $q$ through all dipole-allowed intermediate states. The small component follows as in Equation (8). The phases acquired in the absorption and emission paths (cf. Equation (36)), are given by the corresponding two-photon matrix element and the phase of the photoelectron. In the non-relativistic case, and for photoelectrons with momentum along the common polarization axis of the fields, $\hat{\mathbf{z}}$, they are given as

$$\phi_{\mathrm{a}}^{\mathrm{nrel}} = \arg\left(\sum_\ell M_{\mathrm{a,nrel}}(\ell)e^{i\left(-\ell\frac{\pi}{2} - \sigma_{Z,k,\ell} + \delta_{k,\ell}\right)}Y_{\ell,0}(\hat{\mathbf{z}})\right)$$
$$\phi_{\mathrm{e}}^{\mathrm{nrel}} = \arg\left(\sum_\ell M_{\mathrm{e,nrel}}(\ell)e^{i\left(-\ell\frac{\pi}{2} - \sigma_{Z,k,\ell} + \delta_{k,\ell}\right)}Y_{\ell,0}(\hat{\mathbf{z}})\right), \tag{38}$$

where the short-hand notation

$$M_{a,nrel}(\ell) = M^{(2)}_{nrel}(q, \omega, \Omega_<, b),$$
$$M_{e,nrel}(\ell) = M^{(2)}_{nrel}(q, -\omega, \Omega_>, b) \tag{39}$$

has been used and the subscripts a and e stand for IR absorption and emission, respectively. For photoelectron emission along the $\hat{\mathbf{z}}$-direction, i.e., $\theta = 0$, the spherical harmonic is non-zero only for azimutal quantum number $m_\ell = 0$. The atomic delay, defined as the phase difference divided by $2\omega$, can subsequently be calculated as

$$\tau_A = \frac{\phi_e - \phi_a}{2\omega}. \tag{40}$$

In the Dirac case, there are two distinct differences. First, a sum over $m = \pm 1/2$ is required, because both spin-directions contribute to the emission along the $\hat{\mathbf{z}}$-direction. Second, due to the multi-component wave function the Dirac case the expression gets more involved:

$$\tau_A = \frac{1}{2\omega} \arg \left[ \sum_{m=\pm\frac{1}{2}} \right.$$
$$\left( \sum_\kappa M_a \begin{pmatrix} \chi_{\kappa m}(\hat{\mathbf{z}}) \\ i\zeta\chi_{-\kappa m}(\hat{\mathbf{z}}) \end{pmatrix} e^{i\left(-\gamma\frac{\pi}{2} - \tilde{\sigma}_{Z,k,\gamma} + \nu + \tilde{\delta}_{Z,k,\gamma}\right)} \right)^\dagger$$
$$\times \left. \left( \sum_{\kappa'} M_e \begin{pmatrix} \chi_{\kappa' m}(\hat{\mathbf{z}}) \\ i\zeta\chi_{-\kappa' m}(\hat{\mathbf{z}}) \end{pmatrix} e^{i\left(-\gamma'\frac{\pi}{2} - \tilde{\sigma}_{Z,k,\gamma'} + \nu + \tilde{\delta}_{Z,k,\gamma'}\right)} \right) \right], \tag{41}$$

where the sum over $m = \pm 1/2$ is done incoherently (see e.g., the discussion in ref. [85]) , whereas the sum over $\kappa$ is done coherently. The two incoherent contributions to the atomic delay are due to unresolved photoelectron spin in the final state. The expression for the Wigner and atomic delay for electrons detected along an arbitrary direction have been discussed in ref. [86]

## 4. Method

In the following, we label the full four-component "perturbed wave function", associated with absorption of one photon with angular frequency $\Omega$ and a hole in orbital $b$, by $\left| \rho_{\Omega,b} \right\rangle$, including both radial and spin-angular parts implicitly. As in [41–43], we use here the RPAE-approximation for the many-body response to the absorption of an XUV-photon, albeit within a relativistic framework.

### 4.1. The Form of the Light–Matter Interaction

The standard expression for light–matter interaction,

$$h_I = ec\boldsymbol{\alpha} \cdot \mathbf{A}(\mathbf{r}, t), \tag{42}$$

comes from applying minimal coupling: $\mathbf{p} \to \mathbf{p} + e\mathbf{A}$ to the Dirac Hamilonian in Equation (1). Within the dipole approximation, the vector potential is assumed to be space-independent: $\mathbf{A}(\mathbf{r}, \mathbf{t}) \to \mathbf{A}(\mathbf{t})$. This is often referred to as the "velocity gauge" expression for light—matter interaction:

$$h_I^{velocity} = ec\boldsymbol{\alpha} \cdot \mathbf{A}(t). \tag{43}$$

A unitary transformation can be made to recast the interaction in the alternative "length gauge" form

$$h_I^{\text{length}} = e\mathbf{r} \cdot \mathbf{E}. \tag{44}$$

For details see e.g., ref. [87]. Because our interest here is low-energy photoelectrons, we will stick to the dipole approximation. It is well known that the two gauge forms give identical results when evaluated by an exact wave function, but also for approximations that employ a local potential to describe electron–electron interaction. The non-local exchange potential in the Hartree–Fock approximation can lead to different results in the two gauges when static orbitals are assumed [65,66]. As was shown in the 1970s, the gauge invariance for one-photon processes is restored by the RPAE class of many-body effects [88]. Recently, this was discussed in connection with the calculation of two-photon processes, as needed for the calculation of atomic delays [44], and it was shown that gauge invariance required a full two-photon RPAE treatment. Because ref. [44] also showed that the length gauge results are completely dominated by the time-order where the XUV photon is absorbed first and much less sensitive to final state interactions (after absorption of two photons) than velocity gauge, only the length form will be used here.

With linearly polarized light, we may now write the lowest order approximation of the transition matrix elements from Equation (8) as

$$M^{(1)}(q, \Omega, b) = \langle q \mid ez \mid b \rangle E_\Omega, \tag{45}$$

and similarly the two-photon matrix element in Equation (37) as

$$M^{(2)}(q, \omega, \Omega, b) = \lim_{\xi \to 0^+} \int\!\!\!\!\!\!\!\sum_p \frac{\langle q \mid ez \mid p \rangle \langle p \mid ez \mid b \rangle}{\epsilon_b + \hbar\Omega - \epsilon_p + i\xi} E_\omega E_\Omega, \tag{46}$$

where intermediate states, $p$, are to be summed and integrated over for the bound and continuum part of the spectrum respectively. An important difference compared to the one-photon matrix element is that the two-photon matrix element is intrinsically complex for the above threshold ionization, i.e., when $\hbar\Omega$ exceeds the binding energy, even if correlation effects are neglected.

*4.2. Diagrammatic Perturbation Theory*

The approximation is illustrated by the diagrams in Figure 1, and a detailed derivation can be found in ref. [44]. The solution of the RPAE equations is done iteratively as indicated in Figure 1 and includes the linear response to the interaction with the XUV photon,

$$\left(\epsilon_b \pm \hbar\Omega - h_\kappa^{DF}\right) \mid \rho_{\Omega,b}^\pm \rangle = \sum_p^{exc} \mid p \rangle \langle p \mid \left(d_{\Omega_j} + \delta u_\Omega^\pm\right) \mid b \rangle, \tag{47}$$

where $\delta u_\Omega^\pm$ is the (linearized) corrections to the Dirac–Fock potential induced by the electromagnetic field (cf. Figure 1c–f,i–l). The Dirac–Fock potential, cf. Equation (4) is defined from its matrix element between orbitals $m, n$ (occupied or unoccupied):

$$\langle m \mid u_{DF} \mid n \rangle = \sum_c^{core} \langle \{mc\} \mid V_{12} \mid \{nc\} \rangle, \tag{48}$$

where curly brackets denote anti-symmetrization. $V_{12}$ denotes here the Coulomb interaction. It is also possible to define a Hartree–Fock type potential for the Breit interaction [89,90], but this aspect of the electron–electron interaction is neglected here. In addition to the Dirac–Fock potential, we usually add a so-called projected potential, $u_{proj}$, to the Hamiltonian in Equation (4). Aiming for a final state with a hole in one of the originally occupied orbitals, the projected potential cancels the removed electron's monopole interaction with

all unoccupied orbitals, without affecting the interaction between the electrons in the ground state. More details can be found in [44]. Through this extra potential, some of the contributions from Figure 1c, precisely those which ensure that the photoelectron feels the correct long-range potential, are accounted for already in lowest order. When converged, the iterative procedure gives the same results if the projected potential is used or not, but the convergence is often much improved in the latter case, especially close to ionization thresholds.

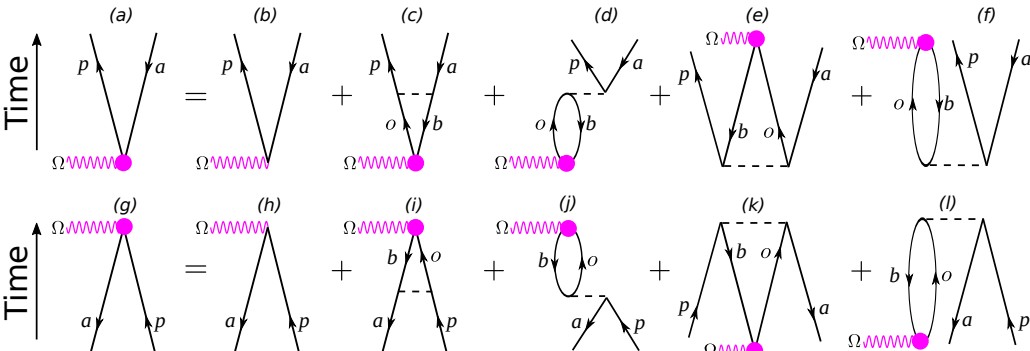

**Figure 1.** Goldstone diagrams illustrating RPAE for the many-body screening of the photon interaction. The set labelled with (**a**,**g**) are forward and backward propagation, respectively. Diagrams (**b**,**h**) are the lowest order contributions, while (**c**–**f**) and (**i**–**l**) give the many-body response. The sphere indicates the correlated interaction to infinite order. The wavy line indicates the photon interaction and the dashed line the Coulomb interaction. Downward lines (labelled with a, b) stand for holes created when electrons are removed from initially occupied orbitals, and upward lines (labelled with o, p) for initially unoccupied orbitals.

The calculations are performed with a basis set obtained through diagonalization of the radial one-particle Dirac–Fock Hamiltonians in a primitive basis of B-splines [91], defined on a knot sequence in a spherical box. B-splines are piecewise polynomials of a given order $k$. The radial components $f$ and $g$ of the relativistic wave function, (cf. Equation (5)) are expanded in B-splines of different orders: typically $k = 7$ and $k = 8$ respectively. It has been shown by Froese, Fischer, and Zatsarinny [92] that the use of different B-spline orders is a way to get rid of the so-called spurious states, which are known to appear in the numerical spectrum after discretization of the Dirac Hamiltonian. Details of the use of B-splines to solve the Dirac equation can be found in ref. [93].

We use further exterior complex scaling (ECS)

$$r \to \begin{cases} r, & 0 < r < R_C \\ R_C + (r - R_C)e^{i\varphi}, & r > R_C, \end{cases} \tag{49}$$

and thus the eigenenergies of the virtual orbitals are complex in general. As a consequence, the energy integration path avoids the pole in Equation (46) and thus the sum and integration over unoccupied states $p$ can be represented by a finite sum [41,42].

With converged RPAE results the two-photon matrix elements in Equation (37) can be calculated for the absorption as well as the emission path to sideband $n$:

$$M^{(2)}_{a/e} = \langle q \mid ez \mid \rho^+_{(2n\mp1)\omega,b} \rangle, \text{ where } \epsilon_q = \epsilon_b + 2n\omega, \tag{50}$$

where length gauge has been assumed. The integration in Equation (50) involves two continuum functions and will not converge on any finite interval. The integrand is therefore divided into two parts. The first is an inner region $0 \leq r < R < R_C$, where the perturbed wave function and final state can be determined numerically on the B-spline basis. The second is an outer region $R \leq r < \infty$ where the functions can be assumed to be solutions to the pure Coulomb problem, albeit with a possible phase shift. By using different breakpoints

$R$, we can check that the result is independent on where the change from numerical to the analytical integration is done. This procedure was described in ref. [42] but has to be slightly changed for the relativistic case, as will be discussed in the next subsection.

### 4.3. The Continuum–Continuum Transition

To evaluate Equation (50), we need the final continuum state, $q$, for a photoelectron of energy $\epsilon_q$, obtained in a relativistic framework and with the phase shift it gets from the many-body environment. A good approximation is found as the solution of

$$h\psi_q = \epsilon_q \psi_q \tag{51}$$

where $h = h_\kappa^{DF} + u_{proj}$. By expanding the radial functions $f$ and $g$ (cf. Equation (5)), in B-splines, $f_q(r) = \sum c_i B_i(r)$, and vice versa for $g$, we can reformulate Equation (51) to a system of linear equations for the coefficients $c_i$. Exclusion of the first B-spline yields a regular solution, that is zero at the origin. This determines $\psi_q$ up to a normalization constant. After normalization, which will be discussed below, $\psi_q$ is used for the first part of the integration in Equation (50), i.e., from zero to $R$. We note in passing that in practice it is enough to obtain the large component of the relativistic wave function for a specific energy, because in the region dominated by the Coulomb potential, the Dirac equation gives the small component directly from the large one:

$$u_g(r) = \frac{c\hbar\left(\frac{d}{dr} + \frac{\kappa}{r}\right)u_f(r)}{\epsilon + 2mc^2 + \frac{e^2 Z_{eff}}{4\pi\varepsilon_0}\frac{1}{r}}, \tag{52}$$

where $Z_{eff}$ is the effective Coulomb potential felt by the escaping electron.

For the second part of the integration, from $R$ to infinity, we need to extract information from the numerical representations of $q$ and $\rho$ to perform the rest of the integral in Equation (50) analytically as was described in ref. [42]. The final state $q$, is a phase shifted regular solution to the Coulomb problem, which should be correctly normalized, and the perturbed wave function $\rho$, is a phase-shifted outgoing solution with an amplitude determined by the photoionization process.

The outgoing solution $\rho$ well outside the ionic core can easily be compared with the pure Coulomb solutions, Equations (23) and (24), combined as in Equation (31), to determine the phase shift, $\tilde{\delta}$ in Equation (25). It can easily be checked that the obtained phase shift is independent of $r$, and then Equations (23) and (24) can be used again to construct the solution at any large $r$.

The final state phase-shifted regular solution from Equation (51) can be complimented by its irregular counterpart through Equation (20), evaluated with the relativistic forms of $\bar{f}$ and $\bar{g}$, and then again the phase shift can be determined from comparision with Equations (23) and (24), combined as in Equation (31), and finally Equation (23) can be used to construct the final state at any $r$.

An additional advantage with the possibility to complement a regular solution with its corresponding irregular solution, and be able to construct the outgoing function, is that it is easy to normalize. The probability flux through the surface of a sphere of radius $R$ is

$$\mathcal{J}(R) = ic\left(u^f(r)^* u^g(r) - u^g(r)^* u^f(r)\right)_{r=R} \tag{53}$$

and is constant for any large value of $R$, far outside the core. Because the asymptotic expressions for the large and small components are simple oscillating waves and their relation is $\zeta$ (cf. Equation (8)), the rate should be $2c\zeta|A|^2$ and from that we can determine the amplitude $A$. From the expression for $\zeta$ in Equation (9), we note the close resemblance with the non-relativistic rate $\hbar k|A|^2/m$, just slightly adjusted for relativistic effects.

The last part of the integral, from $R$ to $r \to \infty$, in Equation (50) can now be calculated as was described in ref. [42], but now with continuum solutions obtained from Equations (23) and (24).

## 5. Results

The two-photon matrix elements for the absorption, $M_a$, and emission, $M_e$, paths are calculated as indicated in Equation (50) and then the atomic delay for electrons emitted in the direction of the laser field polarization is obtained from Equation (41). The Wigner delays are calculated as in Equation (34).

### 5.1. A Light Element: Argon

Results for ionization of argon atoms to the outermost $p$ doublet $3p_{1/2}^{-1}$ and $3p_{3/2}^{-1}$, are shown in Figure 2. The two curves for the atomic delays are, more or less, indistinguishable. The negative atomic delay peak at 50 eV is due to the $3p$-Cooper minimum in the cross section of argon. A slight shift of the negative atomic delays peaks of a few meV is observed. The similarity of the two fine-structure split channels is expected for such a light system with $\Delta E_{FS}^{Ar:3p_j} = 0.18$ meV. The Wigner delays from the two fine-structure channels are also mostly indistinguishable. Just below the threshold for release from the $3s$-orbital, $\sim 30$ eV, there are narrow resonances that are not fully resolved in the present calculation.

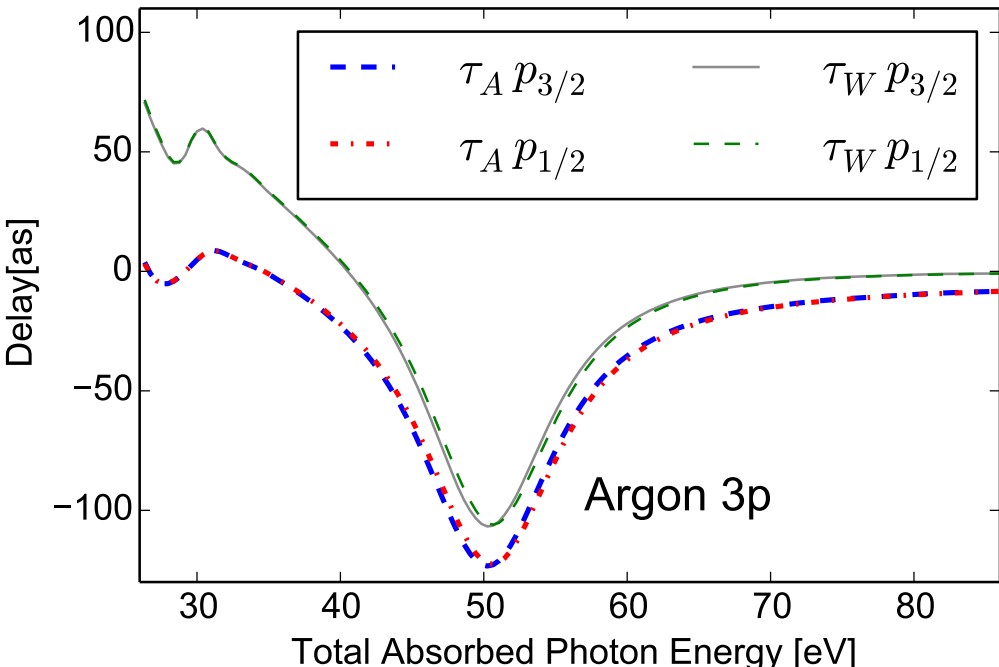

**Figure 2.** The atomic and Wigner delay calculated in length gauge for ionization from Ar $3p_j$, for electrons emitted along the polarization axis. The figure shows the region in the vicinity of the Cooper minimum. The thick dashed blue line shows the atomic delay for electrons ionized from $3p_{3/2}$. It is hardly distinguishable from the dashed–dotted red line that shows the atomic delay for electrons ionized from $3p_{1/2}$. The thin dashed green and solid grey lines show the Wigner delay for electrons ionized from $3p_{1/2}$ and $3p_{3/2}$ respectively. Dirac–Fock orbital energies have been replaced with experimental ionization energies: For $3p_{3/2}$ the binding energy is 15.76 eV, and for $3p_{1/2}$ it is 15.94 eV.

### 5.2. Heavy Elements: Krypton and Xenon

Atomic and Wigner delays for ionization to the outermost $p$-doublet in krypton and xenon are shown in Figures 3 and 4, respectively. Here the delay differences between the electrons ionized to the $4p_{1/2}^{-1}$ and $4p_{3/2}^{-1}$ ($5p_{1/2}^{-1}$ and $5p_{3/2}^{-1}$) in krypton (xenon) show that relativistic effects are important. Differences between the delays are clearly visible on the order of a few eV at the Cooper minima. Such shifts can be expected because the fine-structure shifts are $\Delta E_{FS}^{4p_j} = 0.67$ eV for krypton ($\Delta E_{FS}^{5p_j} = 1.3$ eV for xenon). Furthermore, a difference between the doublet channels is observed at low energies, where xenon shows the largest delay difference that exceeds 10 as.

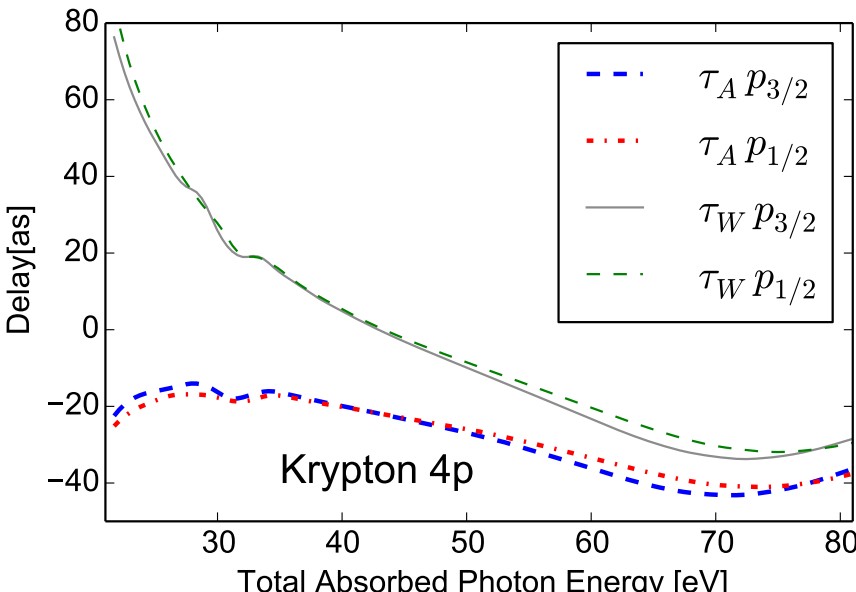

**Figure 3.** The atomic and Wigner delay calculated in length gauge for ionization from Kr $4p_j$, for electrons emitted along the polarization axis. The thick dashed blue line shows the atomic delay for electrons ionized from $4p_{3/2}$, and the dotted–dashed red line shows the atomic delay for electrons ionized from $4p_{1/2}$. The thin dashed green and solid grey lines show the Wigner delay for electrons ionized from $4p_{1/2}$ and $4p_{3/2}$ respectively. Dirac–Fock orbital energies have been replaced with experimental ionization energies: For $4p_{3/2}$ the binding energy is 14.00 eV, and for $4p_{1/2}$ it is 14.67 eV [94]. Dirac–Fock orbital energies are used for the deeper lying orbitals.

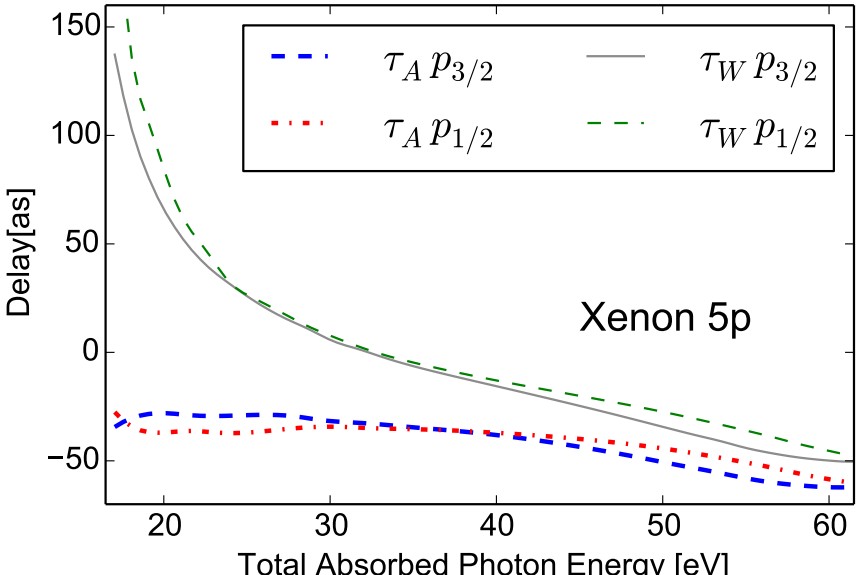

**Figure 4.** The atomic delay calculated in length gauge for ionization from Xe $5p_j$, for electrons emitted along the polarization axis. The thick dashed blue line shows the atomic delay for electrons ionized from $5p_{3/2}$, and the dotted–dashed red line shows the atomic delay for electrons ionized from $5p_{1/2}$. The thin dashed green and solid grey lines show the Wigner delay for electrons ionized from $5p_{1/2}$ and $5p_{3/2}$ respectively. Dirac–Fock orbital energies have been replaced with experimental ionization energies. For $5p_{3/2}$, the binding energy is 12.13 eV, and for $5p_{1/2}$ it is 13.44 eV [94]. Dirac–Fock orbital energies are used for the deeper lying orbitals.

### 5.3. Study of Continuum–Continuum Delay

The difference between the atomic and the Wigner delay is plotted for argon, krypton, xenon, and radon in Figure 5. For all the elements, and all fine-structure components, the results are very similar. This is in accordance with earlier findings, using non-relativistic calculations [41,42,44], and the corresponding numerical continuum–continuum delay: $\tau_{cc}^{\text{MBPT}}$, is shown as a dotted line in Figure 5 for comparison with the relativistic results. Thus, the contribution from the second photon depends on the kinetic energy and the long-range potential, but only weakly, or not at all, on the structure of the remaining ion, or its angular momentum, for photoelectrons emitted along the polarization axis.

Only in the vicinity of Cooper minima, or close to resonances, is there are a deviation from this general trend. We stress that non-relativistic deviations, of a few attoseconds, have also been found for Ar3$p$ at the Cooper minimum by using the effective one-body potential for the final state [44]. In that case, however, the complete 2P2C-RPAE method was used to show that these deviations could be reduced, as shown Figure 9b of ref. [44]. Thus, we may speculate that the present relativistic deviations at the Ar3$p_j$ Cooper minima could be reduced by turning to 2P2C-RRRA theory. On the other hand, the correlation-induced 3$s$-minimum was shown to be non-separable by using the 2P2C-RPAE method, as shown in Figure 9a of [44]. Obviously, fast photoelectrons are also well described by the analytical cc-delay in ref. [33], but more importantly, Figure 5 demonstrates that a universal behaviour extends to much lower energies than expected from the asymptotic theory ($>$20 eV) [33], in good agreement with non-relativistic 2P2C-RPAE matrix elements [44].

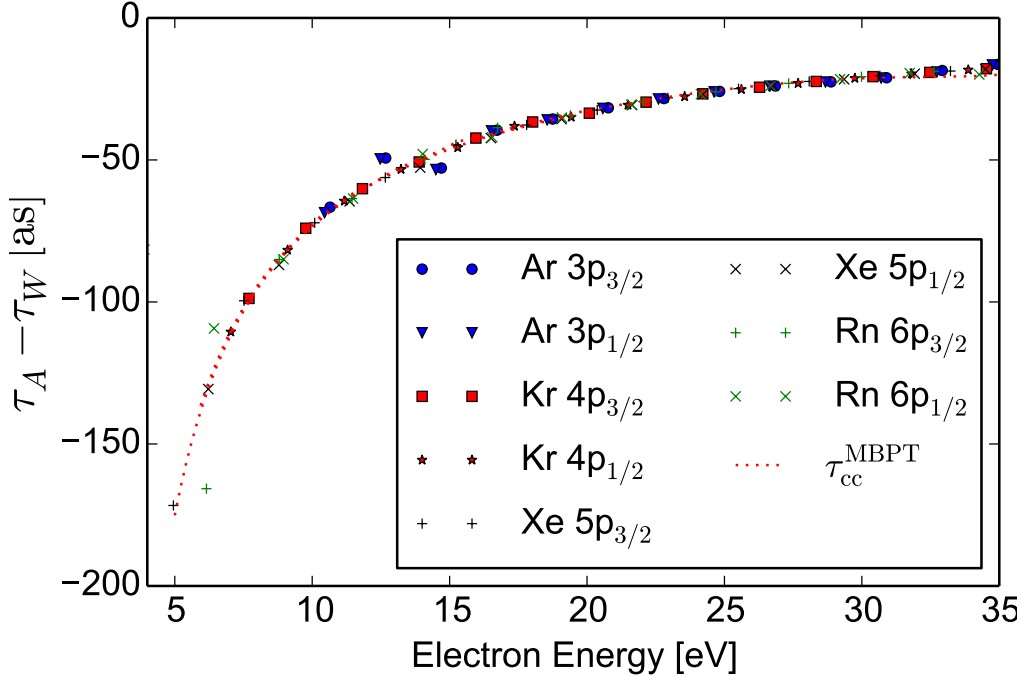

**Figure 5.** The difference between the atomic delay and the Wigner delay for the two outermost orbitals in Ar, Kr, Xe, and Rn calculated in length gauge and for for electrons emitted along the polarization axis. The red dotted line shows the non-relativistic result calculated for Ne 2$p$, i.e., the numerically obtained continuum–continuum delay discussed in the introduction.

### 5.4. Comparison with Experiments

The delay difference between photoelectrons originating from the outermost $p_{3/2}$ and $p_{1/2}$ orbitals in krypton and xenon have been studied in refs. [71,72] by using the interferometric RABBITT technique. In Figure 6, this difference, as calculated here, is shown for xenon. The experiment from ref. [71] includes one data point at 18.6 eV and one at 24.8 eV which are in qualitative agreement both with the calculation presented here, and with accompanying calculations in ref. [71], based on the Wigner delay from RRPA

augmented with the the cc-delay from ref. [33]. Three other data points, on the other hand, differ markedly from both theoretical results. Especially striking are the large measured delays for higher energies (around 30 as at 30 eV), where the calculated result is very small. This might be due to resonances, not fully accounted for in the calculations, as discussed in ref. [71].

Moreover, a higher energy region has been explored. Ref. [72] measured the the delay difference for the xenon $5p$ fine-structure components for the sideband at 90 eV (with IR photon energies of 1.55 eV) to $\tau_A(5p_{3/2}) - \tau_A(5p_{1/2}) = 14.5 \pm 9.3$ as. Moreover, the calculated delay is much smaller, around 2 as (not shown in the figures). We note that the cross section to produce photoelectrons in the $5\ell_j$ channels at around 90 eV photon energy is comparable to those for $4d$ and shake-up satellites [95]. Because shake-up channels can have significantly larger delays [12], this region might need a more careful investigation of all competing channels.

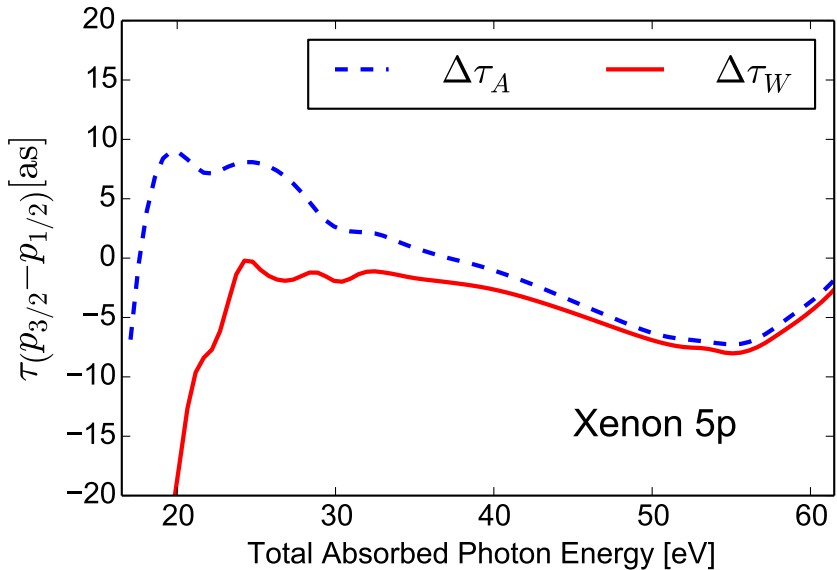

**Figure 6.** The delay difference between photoelectrons originating from the $5p_{3/2}$ and $5p_{1/2}$ orbitals in xenon. The dashed blue line shows the atomic delay, and the solid red the Wigner delay.

Figure 7 shows finally the atomic and Wigner delay for photoelectrons released from the xenon $4d$ orbitals. The result agrees within error bars with the measurement, from threshold up to ~100 eV, in ref. [74]. It is interesting to note the large difference between the two channels, defined by the two fine-structure components, in the region just above the $4d$ thresholds at 67.5 and 69.5 eV, and the rapid variation of the delay with photon energy. The experiments in refs. [96,97] have shown that also the cross-section branching ratio (for leaving the ion with $4d_{3/2}^{-1}$ or $4d_{5/2}^{-1}$) shows a rapid variation in this region. In both cases, this behaviour can be traced back to the presence of two resonances close to threshold. They are of $^3$D and $^3$P character and cannot be populated by one-photon absorption in a non-relativistic description. The spin-orbit-induced singlet-triplet mixing opens, however, the route to ionization through these resonances, and thus for a population transfer from one final channel to the other. This has been further discussed in refs. [74,98]. We note that although the resonances in argon, mentioned above, are just unresolved in the calculation, the reason that these xenon resonances are not seen directly is not a question of resolution. The cross section in this region is completely dominated by the so-called giant resonance of $^1$P character and the spin-orbit-induced resonances can simply not be seen against this background. Still, their mark in the more sensitive observables, such as atomic delays and branching ratios, is clearly seen.

Also for xenon $4d$ ref. [72] gives a value at 90 eV: $\tau_A(4d_{5/2}) - \tau_A(4d_{3/2}) = -4.0 \pm 4.1$ as, which agrees with our value of $-1.2$ as.

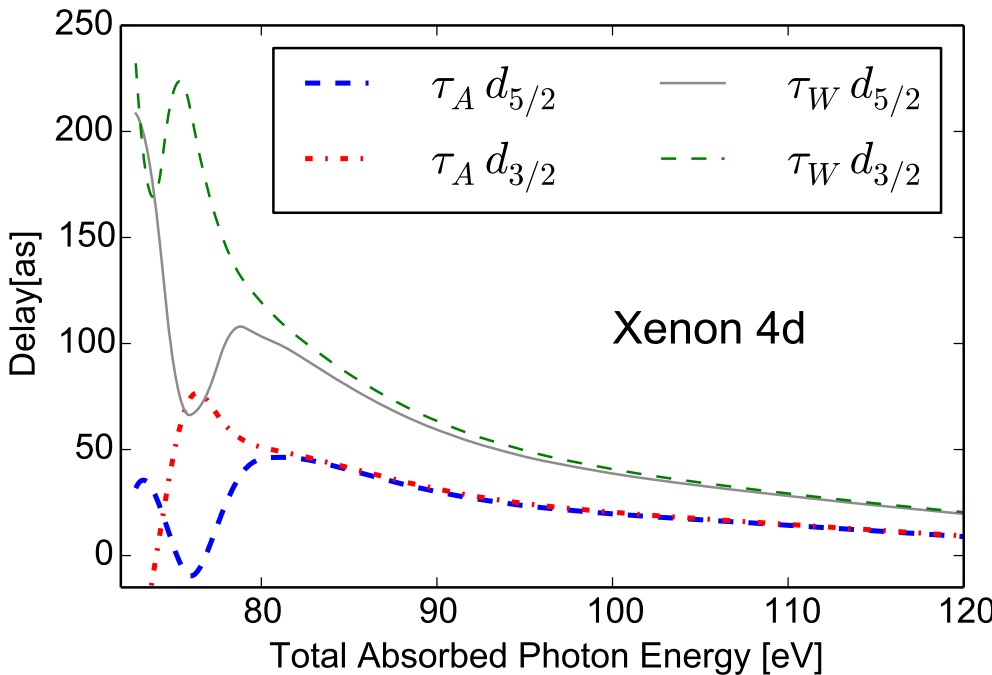

**Figure 7.** The thick dashed blue line shows the atomic delay for electrons ionized from $4d_{3/2}$, and the dotted–dashed red line shows the atomic delay for electrons ionized from $4d_{5/2}$. The thin dashed green and solid grey lines show the Wigner delay for electrons ionized from $4d_{3/2}$ and $4d_{5/2}$ respectively. Dirac–Fock orbital energies have been replaced with experimental ionization energies. For $4d_{5/2}$ the binding energy is 67.5 eV, and for $4d_{3/2}$ it is 69.5 eV [99].

## 6. Conclusions

We have shown how two-photon above-threshold ionization can be treated in a relativistic framework from first principles. Correlation is included in the photoionization process at the level of the relativistic random phase approximation. As in the non-relativistic case, the calculation of the subsequent continuum–continuum transition relies on knowledge of the form of the intermediate wave packet when it is well outside the atomic core. For this purpose, we present a convenient recursive formula for both the large and small component of the regular and irregular solution to the relativistic Coulomb problem. The procedure have been applied to a few heavy elements, and it is shown that the separation of the atomic delay into a Wigner–Smith–Eisenbud delay and a universal continuum–continuum works reasonably well also for these systems.

We have further demonstrated qualitative agreement with existing experimental photoionization-delay data for ionization from the $4d$-orbitals in xenon, and with lower energy results from the outermost orbitals in xenon and krypton. For higher photon energies, experiments report considerably larger delay differences between the fine-structure split channels than supported by the calculations. This might be connected to resonances or interfering shake-up channels, which can hopefully be resolved in the future.

**Author Contributions:** Conceptualization, E.L. and J.M.D.; methodology, E.L., J.M.D. and J.V.; software, J.V. and E.L.; validation, E.L., J.M.D. and J.V.; formal analysis, E.L., J.M.D. and J.V.; investigation, J.V.; resources, E.L. and J.M.D.; data curation, J.V.; writing—original draft preparation, E.L.; writing—review and editing, E.L. and J.M.D.; visualization, J.V. and E.L.; supervision, E.L. and J.M.D.; project administration, E.L. and J.M.D.; funding acquisition, E.L. and J.M.D. All authors have read and agreed to the published version of the manuscript.

**Funding:** This research was funded by the Knut and Alice Wallenberg Foundation: Grant No. 2017.0104 and 2019.0154, from the Swedish Research Council: GrantNo. 2018-03845 and 2020-03315, and from the Olle Engkvist Foundation: Grant No. 194-0734.

**Institutional Review Board Statement:** Not applicable.

**Informed Consent Statement:** Not applicable.

**Data Availability Statement:** Not applicable.

**Conflicts of Interest:** The authors declare no conflict of interest.

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
