# Peer review of "Relativistic Two-Photon Matrix Elements for Attosecond Delays"

_atoms, doi:10.3390/atoms10030080_

Round 1

Reviewer 1 Report

The manuscript “Relativistic two-photon matrix elements for attosecond delays” by J. Vinbladh et al presets very substantial theoretical analysis of two-photon amplitudes involving continuum part of atomic spectra. The authors pay main attention to relativistic effects.

Calculations of these matrix elements are essential part of RABBIT investigations as well as other processes involving continuum, and therefore the manuscript may be interesting for many theoreticians. In spite of the manuscript is quite long, it is well-written and easy to read. Apparently I will recommend this paper to my PhD students.

In my opinion the manuscript is suitable for “atoms” and may be published at present form

Author Response

We thank the referee for the positive comments.

Reviewer 2 Report

Title: Relativistic two-photon matrix elements for attosecond delays

            The manuscript presents the theory of one-photon ionization and two-photon above-threshold ionization, which was formulated for applications to heavy atoms in attosecond science using the Dirac-Fock formalism. Moreover, a direct comparison of the Wigner-Smith-Eisenbud delays for photoionization has been made with delays from the Reconstruction of Attosecond Beating By Interference of Two-photon Transitions (RABBIT) method. The authors have shown how two-photon above-threshold ionization can be treated in a relativistic framework from first principles. They have demonstrated qualitative agreement with existing experimental photoionization-delay data for ionization from the 4d-orbitals in xenon, and with lower energy results from the outermost orbitals in xenon and krypton.

The manuscript is very well written and the results are interesting, and presented in clear and accessible way. The literature is surprisingly rich. The paper evidently contributes to the progress in the field and the importance of the research is clearly demonstrated in the manuscript.

In conclusion the investigated in the manuscript matter matches the issues presented in Atoms, therefore I recommend the paper for publication in Atoms journal without any further corrections.

Author Response

(The authors gave the same response as above.)

Reviewer 3 Report

This is an excellent paper dealing with the relativistic theory of two photon photoabsorption and attosecond time delay.  It is well-written and well-referenced. The only suggestion I have for improvement is to enlarge equation 3 to actually add the eigenvalue equation there including E or epsilon (or both) for clarity.

Author Response

We thank the referee for the positive comments and have followed his/her advice and  added the eigenvalue to Eq. 3